# Evaluation of Gasoline Evaporative Emissions from Fuel-Cap Removal after a Real-World Driving Event

**Hiroo Hata** [1,*] **, Syun-ya Tanaka** [2] **, Genta Noumura** [2] **, Hiroyuki Yamada** [3] **and Kenichi Tonokura** [2,*]

[1] Tokyo Metropolitan Research Institute for Environmental Protection, 1-7-5 Sinsuna, Koto-ku, Tokyo 136-0075, Japan

[2] Department of Environment Systems, Graduate School of Frontier Sciences, The University of Tokyo, 5-1-5 Kashiwanoha, Kashiwa, Chiba 277-8563, Japan; 1463751149@edu.k.u-tokyo.ac.jp (S.-y.T.); 9815352990@edu.k.u-tokyo.ac.jp (G.N.)

[3] Department of Mechanical Engineering, Tokyo Denki University, 5 Senjyu-Asahimachi, Adachi-ku, Tokyo 120-8551, Japan; h-yamada@mail.dendai.ac.jp

* Correspondence: hata-h@tokyokankyo.jp (H.H.); tonokura@k.u-tokyo.ac.jp (K.T.)

**Abstract:** This study evaluated gasoline evaporative emissions from fuel-cap removal during the refueling process (or "puff loss") for one gasoline vehicle in the Japanese market. Specifically, the puff loss emissions were measured after a real-world driving event in urban Tokyo, Japan for different seasons and gasoline types. The experimental results indicated higher puff loss emissions during summer than in winter and spring despite using low vapor pressure gasoline during summer. These higher puff loss emissions accounted maximally for more than 4 g of the emissions from the tested vehicle. The irregular emission trends could be attributed to the complex relationships between physical parameters such as fuel-tank filling, ambient temperature, ambient pressure, and gasoline vapor pressure. Furthermore, an estimation model was developed based on the theory of thermodynamics to determine puff loss emissions under arbitrary environmental conditions. The estimation model included no fitting parameter and was in good agreement with the measured puff loss emissions. Finally, a sensitivity analysis was conducted to elucidate the effects of three physical parameters, i.e., fuel tank-filling, ambient pressure, and gasoline type, on puff loss emissions. The results indicated that fuel tank-filling was the most important parameter affecting the quantity of puff loss emissions. Further, the proposed puff loss estimation model is likely to aid the evaluation of future volatile organic compound emission inventories.

**Keywords:** puff loss; VOCs; gasoline vehicle; evaporative emission; refueling; thermodynamics

## 1. Introduction

Volatile organic compounds (VOCs) are the precursors of two global toxic air pollutants, i.e., tropospheric ozone and secondary organic aerosol-related fine particulate matter ($PM_{2.5}$). Tropospheric ozone and $PM_{2.5}$ have been shown to be significantly detrimental to human health and are associated with increased premature mortality in humans [1–5]. Although these pollutants are associated with respiratory diseases and asthma in humans, detailed mechanisms elucidating their effects on human physiology and health are lacking [6]. Regardless, the toxicity of these two pollutants is well established [1–6]. Consequently, several global environmental policies have focused on the mitigation of ozone and $PM_{2.5}$. For example, the Japanese environmental standard for tropospheric ozone corresponds to hourly values not exceeding 0.06 ppm. Similarly, the Japanese environmental standard for $PM_{2.5}$ corresponds to average yearly and daily values of less than 15 and 35 $\mu g/m^3$,

respectively [7]. Notably, ~90% of the monitoring stations in Japan met the environmental standard for $PM_{2.5}$ while none of the monitoring stations met the environmental standard for tropospheric ozone (or photo-chemical oxidant) in 2017 [7]. While developed countries such as the United States and European Union countries have air quality levels similar to that in Japan [8,9], middle- and low-income countries including China are still suffering from extreme air pollution and carbon emission, and managing air quality based on the careful planning is still challenging [10,11]. Therefore, for both developed and developing countries, it is necessary to develop mitigation strategies for air pollutants, including ozone precursors.

Several anthropogenic sources, including stationary and mobile sources, contribute to VOC emissions. Our previous study utilized a chemical transport model to conclude that both light-duty and heavy-duty vehicles significantly contributed to the tropospheric ozone concentration in the Kanto region of Japan, thus indicating a need to mitigate vehicular exhaust emissions [12]. These emissions can be classified as tailpipe and evaporative emissions. Tailpipe emissions result from fuel combustion in the engine during driving and comprise toxic pollutants such as VOCs, $NO_x$, and CO. Recent catalysis technologies have successfully reduced toxic air pollutants in tailpipe emissions by purifying the exhaust emissions. In this regard, three-way catalysts have been utilized for gasoline emissions [13] while $NO_x$ storage-reduction catalysts and selective catalytic reduction systems have been utilized for diesel emissions [14]. Evaporative emissions result from fuel-based VOC evaporation in gasoline vehicles; further, these emissions can be attributed to running losses, hot-soak losses, diurnal breathing losses, refueling emissions, and puff losses. Previous studies have comprehensively examined the different mechanisms underlying evaporative emissions [15–21]. However, few studies have focused on puff loss emissions, which have been deemed as potentially significant atmospheric pollutants by the US Environmental Protection Agency [22]. Puff loss emissions are evaporative emissions released during the fuel-cap removal step of vehicle refueling. In particular, puff loss emissions have been attributed to fuel tank warming, which accelerates VOC evaporation in the empty space of the fuel tank. To the best of our knowledge, no previous study has comprehensively evaluated puff loss emissions. Therefore, this study experimentally investigated the effects of environmental (including seasonal) conditions, Reid vapor pressure (RVP) of gasoline, and fuel tank-filling on puff loss emissions in one gasoline vehicle. Subsequently, an estimation model was constructed based on the theory of thermodynamics to estimate the puff loss emissions. Furthermore, the estimated results were compared with the experimental results. Finally, a sensitivity analysis was conducted to elucidate the impacts of different environmental parameters on puff loss emissions. Specifically, a theoretical model was proposed to identify the critical physical parameters affecting the puff loss emissions. The findings of this study will aid the accurate evaluation of VOC emission inventories.

## 2. Methodology

### 2.1. Measurement of Puff Loss Emissions

In this study, a gasoline cargo vehicle (official fuel tank volume: 70 L) was selected to evaluate the quantity of puff loss emissions. The detailed specifications of the tested vehicle are presented in Table A1 of Appendix A. Two thermocouples were attached inside the fuel tank to monitor the liquid and gas phase temperatures. Further, one thermocouple was attached to the side door to monitor the ambient temperature. Two types of gasoline were utilized to identify the RVP dependency of puff loss emissions, i.e., summer grade gasoline (SGG, RVP~58 kPa) and winter grade gasoline (WGG, RVP~88 kPa). The gasoline composition was comprehensively analyzed by a gas chromatography flame ionization detector (GC/FID) following the Japanese Industrial Standard K2536-2 [23] conducted by the SVC Tokyo Company. The compositional data for SGG and WGG are presented in Tables A2–A4. Notably, the gasoline composition could not be determined for April 2020 owing to a delivery accident. However, considering the results of our previous study, we presumed that the gasoline used in April 2020 was similar to the WGG used in December 2019 [19]. The test periods were July and

December 2019 and April 2020. The experiments were conducted under real-world driving conditions in Central Tokyo, Japan. The running distance of 6 km was determined from an online questionnaire on the average running distance between the start point and the service station. The fuel cap was covered by a polyethylene bag immediately after the vehicle stopped running. Subsequently, the fuel cap was opened for 30 s to collect the evaporated gas into the bag. Thereafter, the collected gas was vacuumed by a pump (100 L/min) for 1 min and transported to two activated carbon canisters which were connected in series. Most of the gas was trapped by the front canister, while the rear canister was set to collect the gas that had passed through the front canister. The quantity of puff loss emissions (unit: g) was determined as the difference in the weight of the canister before and after the trapping of the evaporation gas. The effect of fuel tank-filling on the quantity of puff loss emissions was determined based on different tank-fill conditions ranging from 10 to 60 L. The fuel tank-filling experiments were conducted immediately after measuring the quantity of puff loss emissions. Table 1 presents a summary of the experimental conditions. Notably, ambient temperature and ambient pressure are important factors affecting the quantity of puff loss emissions (discussed later in Sections 4.1 and 4.2). The data on ambient temperature, ambient pressure, and weather at 12 PM on the test day were obtained from the Japan Meteorological Agency [24] (Table 1).

**Table 1.** Summary of the experimental conditions.

| Parameter | Experimental Date | | | |
|---|---|---|---|---|
| | **30 July 2019** | **17 December 2019** | **24 December 2019** | **16 April 2020** |
| Gasoline type | SGG [1] | SGG [1] | WGG [2] | WGG [2] |
| Tank-filling (L) | 20, 40, 60 | 20, 30, 40, 50 | 10, 20, 30, 40, 50 | 10, 20, 30, 40, 50 |
| Ambient temperature at 12 PM (°C) | 33.0 | 8.3 | 12.2 | 14.8 |
| Ambient pressure at 12 PM (kPa) | 100.8 | 101.5 | 101.8 | 101.4 |
| Weather | Sunny | Cloudy | Sunny | Sunny |

[1] SGG: Summer grade gasoline. [2] WGG: Winter grade gasoline.

### 2.2. Composition of VOCs

The puff loss emissions were sampled by a 10 mL syringe and trapped in a high-vacuumed 6 L stainless steel canister. Thereafter, the trapped puff loss compounds were diluted by pure nitrogen gas to a pressure of 150 kPa. The 76 components in the puff loss emissions were analyzed by a gas chromatography mass spectrometer and flame ionization detector (GC-MS/FID; GCMS-QP2020, Shimadzu Corporation, Kyoto, Japan). A detailed account of the experimental setup is provided in our previous study [25].

### 2.3. Puff Loss Estimation Model

#### 2.3.1. Estimation of VOC Composition in Puff Loss Emissions

Assuming that the thermodynamic condition of liquid gasoline is ideal liquid, the vapor pressure of gasoline ($P_{evap}$) was derived from Raoult's law as indicated in Equation (1),

$$P_{evap} = \sum_i P^i_{evap} x_i \qquad (1)$$

where $P^i_{evap}$ is the vapor pressure of gasoline component $i$ (Pa) and $x_i$ is the molar fraction of gasoline component $i$. $P^i_{evap}$ was estimated following the Antoine equation (Equation (2)). The parameters $A^i$, $B^i$, and $C^i$ for component $i$ were obtained from the NIST Chemistry WebBook [26],

$$\log P^i_{evap} = A^i - \frac{B^i}{T + C^i} \qquad (2)$$

where $T$ is the equilibrium temperature (K) of the gaseous and liquid phases in the fuel tank. Notably, only three Antoine parameters were used in Equation (2) because a few low volatility components were not derived. Considering the limitation of the Antoine parameters, the following Clausius–Clapeyron equation (Equation (3)) was utilized to evaluate $P^i_{evap}$,

$$P^i_{evap} = P^i_r \exp\left\{ \frac{\Delta H^i_{evap}}{R} \left( \frac{1}{T_r} - \frac{1}{T} \right) \right\} \tag{3}$$

where $P_r^i$ is the RVP (kPa), $\Delta H^i_{evap}$ is the evaporation enthalpy for component $i$ (J/mol), $R$ is the gas constant (8.314 J/(K mol)), and $T_r$ is the temperature at 100 °F (310.95 K). The values for $P_r^i$ and $\Delta H^i_{evap}$ were obtained from the NIST Chemistry WebBook [26]. The molar fraction of liquid fuel ($x_i$) was estimated from the fuel component analysis of gasoline mentioned in Section 2.1 and Table A2. In this study, *n*-alkane, *iso*-alkane, alkenes, naphthene, and aromatics with carbon numbers less than 11 were considered to evaluate $P_{evap}$. Notably, components with carbon numbers greater than three are characterized by several isomers. For example, butene has four isomers, i.e., 1-butene, *cis*-2-butene, *trans*-2-butene, and *iso*-butene, based on the order of the carbon double bond. Further, the number of isomers is proportional to the carbon number; thus, it was difficult to account for all isomers in Equations (1) and (2). Therefore, in this study, only one isomer was considered as representative of each composition type. This assumption may result in the erroneous estimation of $P_{evap}$ owing to the differences in the physical properties of the isomers. However, this error was negligibly small for most components. Table S1 summarizes the representative molecules of different component types as well as the physical parameters used in Equations (2) and (3). According to Boyle's law, the ratio of $P^i_{evap}$ corresponds to the composition ratio of puff loss emissions.

### 2.3.2. Estimation of the Quantity of Puff Loss Emissions

Puff loss emissions were generated owing to the pressure difference between the warmed fuel tank and the ambient air. The quantity of puff loss emissions was estimated according to the ideal gas equation as indicated in Equation (4).

$$\Delta w = \frac{M_g V_m}{RT} \Delta P \times \frac{P_{evap}}{P_{total}} \tag{4}$$

where $\Delta w$ is the quantity of puff loss emissions (g), $M_g$ is the mean molecular weight of the tested gasoline (g/mol), $V_m$ is the empty space volume of the fuel tank (L), $T$ is the gas phase temperature inside the fuel tank (K), $\Delta P$ is the pressure difference between the fuel tank and ambient air (kPa), and $P_{total}$ is the total pressure inside the fuel tank after the test run (kPa). $P_{evap}$ in Equation (4) was calculated from Equation (1). Notably, $V_m$ represented the difference between the fuel tank volume and fuel tank-filling. Further, the official fuel tank volume indicated by a car manufacturer is not the real fuel tank volume because the fuel tank is usually designed to contain dead space. The dead-space volume ($V_{ex}$) was not officially disclosed by the car manufacturer; therefore, $V_{ex}$ was treated as a fitting parameter for the experimental results obtained in this study. $P_{total}$ was calculated by the following process. The gas inside the fuel tank was composed of both air and vapor from the liquid fuel. The air pressure inside the fuel tank before the test run ($P^0_{air}$) was calculated using Equation (5).

$$P^0_{air} = P_{amb} - P^0_{evap} \tag{5}$$

where $P_{amb}$ is the ambient pressure (kPa) and $P^0_{evap}$ is the evaporative gas pressure (kPa) inside the fuel tank before the test run. Assuming that the gas phase temperature inside the fuel tank varied from

$T_g{}^0$ to $T_g{}^1$, the air pressure inside the fuel tank after a test run over the specified distance ($P^1{}_{air}$) was derived based on Charles' law.

$$P^1_{air} = \left(\frac{T^1_g}{T^0_g}\right) P^0_{air} \tag{6}$$

Hence, the total pressure inside the fuel tank after the test run ($P_{total}$) was calculated by the following Equation (7).

$$P_{total} = P_{evap} + P^1_{air} \tag{7}$$

Finally, $\Delta P$ was calculated by the following Equation (8).

$$\Delta P = P_{total} - P_{amb} \tag{8}$$

The maximum pressure difference between a fuel tank and ambient air is determined to eradicate events of extremely high vapor pressure. Notably, a check valve opens at the maximum vapor pressure to release the vapor into the atmosphere. In this study, the maximum pressure of the tested vehicle was 4.9 kPa. Consequently, the maximum $\Delta P$ (as defined in Equation (8)) was set to 4.9 kPa. Notably, the maximum $\Delta P$ value is different for different vehicles, and there has been a recent trend of an increasing maximum pressure in fuel tanks to prevent VOCs being emitted into the air and inhaled by humans in the refueling process. Therefore, further research is necessary to construct puff loss estimation models (Equation (4)) for different vehicle models.

## 3. Results and Discussion

### 3.1. Puff Loss Emissions from Real-World Driving

The measured puff loss emissions after real-world driving are presented in Figure 1. Notably, puff loss emissions were generated in all experiments but the experiments of July 2019 with 60 L fuel tank-filling and of December 2019 with 50 L fuel tank-filling. The experimental results indicated that the puff loss emissions in December 2019 (SGG) were lower than those observed for the other experiments. This could be attributed to the low ambient temperature in December and the utilization of a low volatility fuel. Despite this fact, using low volatility fuel at ambient temperatures would lead to the low combustion efficiency of the engine system and, therefore, the combination of low volatility fuel with low ambient temperatures is not recommended. The highest puff loss emissions were observed in April 2020 (WGG). This could be attributed to the relatively high ambient temperature in April and the utilization of a high volatility fuel. Further, the puff loss emissions were largely inversely correlated with fuel tank-filling. However, this trend was inconsistent (e.g., the relationships between puff loss emissions and 10 and 20 L fuel tank-filling in April 2020). Therefore, in addition to fuel tank-filling, the puff loss emissions were also affected by fuel tank temperature and fuel vapor pressure. Consequently, we presumed that the puff loss emissions were affected by multiple factors, thus accounting for their inconsistent relationship with fuel tank-filling. A detailed discussion of these factors is presented in Section 4.2 The experimentally measured puff loss emission values are listed in Table S2 of the Supplementary Materials.

### 3.2. Temporal Profiles of Fuel Tank Temperature

Figure 2 presents the temporal profiles of ambient and fuel tank temperatures for the test runs in summer (July 2019) and winter (December 2019). The fuel tank temperature comprised the liquid gasoline temperature and the temperature of the gas in the empty space of the fuel tank. Notably, the measured temperature values were higher in summer than winter, thus indicating the seasonal dependency of puff loss emissions. As indicated in Figure 2a, the temperatures of liquid gasoline and the fuel tank gas gradually linearly increased with the driving time during summer. Moreover, these temperatures were similar to the ambient temperature at the end of the drive. On the other hand, as indicated in Figure 2b, the temperatures of liquid gasoline and the fuel tank gas were relatively

stable and did not increase rapidly during winter. Notably, the temperature of the fuel tank gas was higher than that of liquid gasoline. This could be attributed to the different specific heats of the two phases. Consequently, the liquid and gas phases were characterized by an incomplete liquid–gas equilibrium. All the temperature values obtained after the experimental measurements are listed in Tables S3–S5 of the Supplementary Materials.

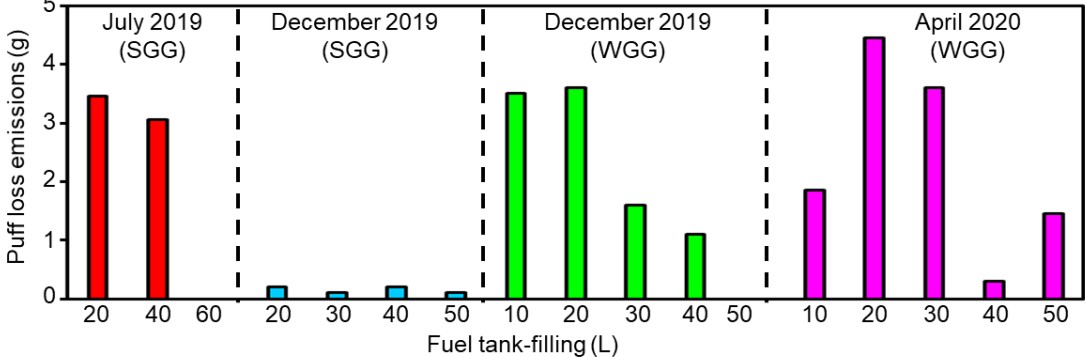

**Figure 1.** Puff loss emissions from real-world driving experiments conducted across three seasons: July 2019, December 2019, and April 2020. SGG and WGG indicate summer grade gasoline and winter grade gasoline, respectively.

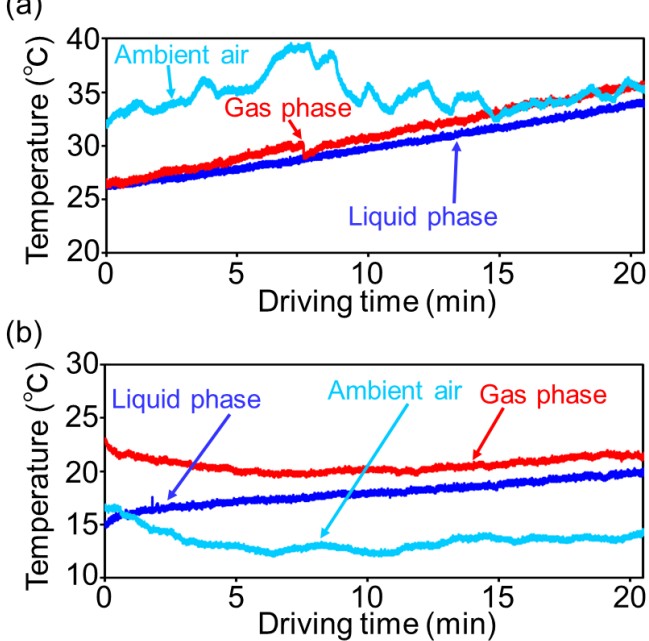

**Figure 2.** Relationships between ambient and fuel tank temperatures and driving time. (**a**) Summer (July 2019) and (**b**) winter (December 2019).

### 3.3. VOC Compositions of Liquid Fuel and Puff Loss Emissions

Figure 3 presents the VOC compositions of liquid fuel and puff loss emissions in December 2019. Notably, the VOC composition of puff loss emissions was determined from Equation (2) (estimated values) as well as from the GC-MS/FID (measured values; Table A5). The results highlighted that WGG indicated a lower ratio of aromatics than SGG to enhance the engine start-up performance in cold temperatures. Consequently, the WGG puff loss emissions were characterized by higher ratios of alkanes and alkenes than the SGG puff loss emissions. Furthermore, the estimated VOC composition of WGG was in good agreement with the experimental results. Contrarily, the estimated VOC

composition of SGG relatively underestimated the experimental results. This could be attributed to the underestimation of vapor pressure by the Antoine Equation (2) and Clausius–Clapeyron Equation (3). Regardless, the overall estimated composition of puff loss emissions was in good agreement with the experimental results. Therefore, we conclude that the VOC composition of puff loss emissions can be evaluated from the composition of liquid gasoline as well as the Antoine Equation (2) and Clausius–Clapeyron Equation (3).

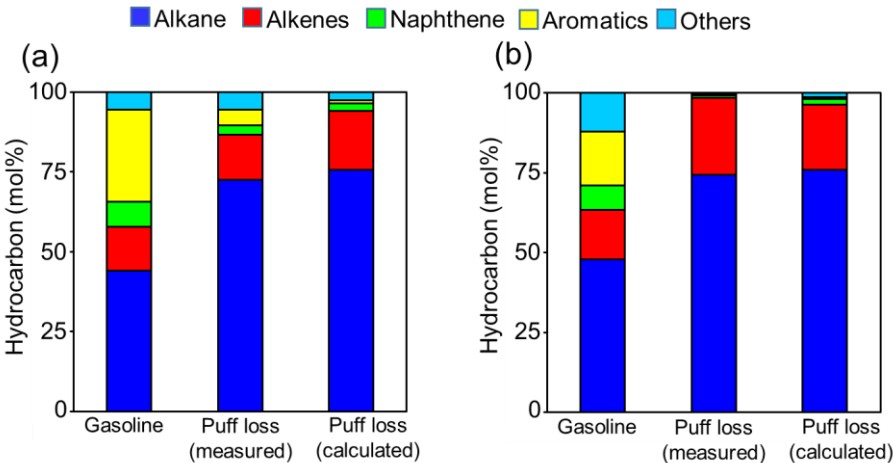

**Figure 3.** The chemical compositions of liquid gasoline and puff loss emissions (measured and calculated values) for (**a**) SGG and (**b**) WGG.

## 4. Puff Loss Estimation Model

### 4.1. Accuracy of the Puff Loss Estimation Model

The puff loss estimation model was based on Equation (4). The measured puff loss emissions under different experimental conditions are presented in Figure 1 while the calculated values are presented in Figure 4. The dead-space volume of the fuel tank ($V_{ex}$) was set to 5 L, of which the puff loss emissions calculated by the estimation model was in good agreement with the experimental results of this study. Since we did not record the exact duration of the experiment and could not determine the exact ambient pressure for each condition, the ambient pressures at 12 p.m. listed in Table 1 were applied in the calculations. The duration of each test was 6 h, from 9:00 a.m. to 3:00 p.m., and according to data provided by the Japan Meteorological Agency, there were no drastic changes in ambient pressure [24]. Therefore, the use of ambient pressure at 12 p.m. in the calculations did not lead to considerable errors. The calculated puff loss emissions were consistent with the experimental results; however, the estimation model relatively overestimated the low emission range of the experimental results. This discrepancy could be attributed to the errors in the experimental methods. In this study, the puff loss emissions were measured based on the weight difference of an activated carbon canister before and after the adsorption of VOCs. We assumed that the puff loss emissions were not completely adsorbed by the activated carbon. This could be attributed to events such as leakage from small unintended vents or VOC adsorption by the bag. The apparent error was also observed for the red vacant plot, of which the experimental result showed ~2 g emission, while the model evaluated more than 4 g emission; this discrepancy can be presumed to be attributed to the same reason of the low emission range. Notably, the estimated and measured puff loss emissions were in good agreement, regardless of the experimental errors. However, the estimation model was fitted to only one vehicle. Therefore, it is important to validate the accuracy of the estimation model for other gasoline vehicles to subsequently evaluate VOC emission inventories.

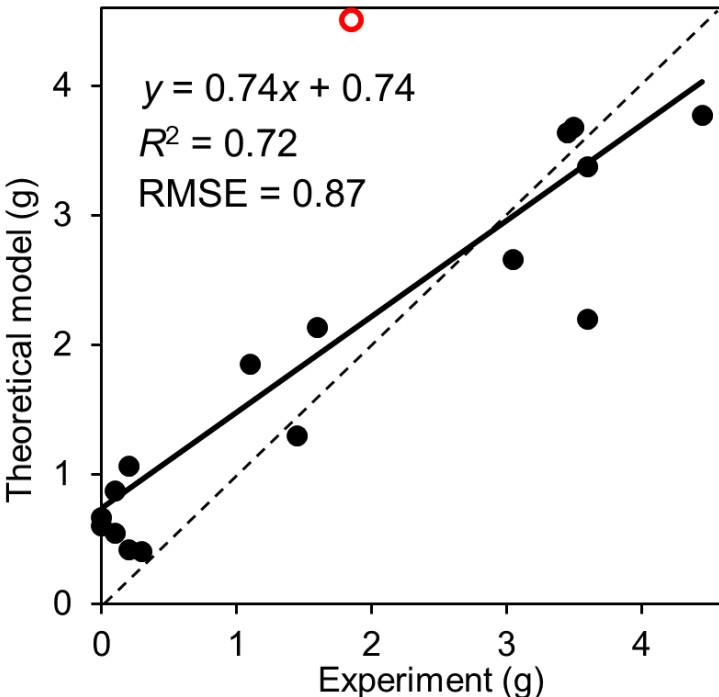

**Figure 4.** Fit between the estimated (Equation (4)) and measured puff loss emissions. Solid line represents the regression line between measured and calculated puff loss emissions. The red vacant circle indicates the plot with serious error between the model calculation and the experimental result.

Figure 5 indicates the pressure difference between the inner fuel tank and the ambient experimental conditions after real-world driving (Equations (5)–(8)). The maximum pressure difference in the tested vehicle was 4.9 kPa. The experimental results indicated that the maximum inner pressure difference was exceeded for the following conditions: fuel tank-filling of 10 and 20 L in July 2019 (SGG), December 2019 (WGG), and April 2020 (WGG) and fuel tank-filling of 40 L in July 2019 (SGG). Therefore, the high puff loss emissions under these experimental conditions (Figure 1) could be attributed to both low fuel tank-filling and high pressure difference between the interior and exterior of the fuel tank.

### 4.2. Sensitivity Analysis of the Puff Loss Estimation Model

In this study, the sensitivity of the puff loss estimation model was determined by altering the parameter values from a base scenario. Specifically, the base scenario comprised the following conditions: fuel tank-filling of 20 L, ambient pressure of 101 kPa, and different gasoline types and seasons (July 2019 (SGG), December 2019 (WGG), and April 2020 (WGG)). The maximum vapor pressure in the fuel tank of the test vehicle was set to 4.9 kPa. Figure 6 presents the sensitivity of the puff loss emissions to each parameter. As indicated in Figure 6a, the puff loss emissions were nearly linearly correlated with fuel tank-filling. In general, refueling was conducted at a fuel tank capacity of 10–30%, thus resulting in approximately 3 to 4 g of puff loss emissions. As indicated in Figure 6b, the puff loss emissions were also linearly correlated with ambient pressure. However, the variation in the maximum and minimum emissions during each season was approximately 10%. Therefore, ambient pressure was not a critical factor affecting the puff loss emissions. The terrestrial altitude in Japan ranges from 0 to 1000 m (average altitude: 68 m) [27], which corresponds to an ambient pressure of ~90–101 kPa. Thus, altitude does not significantly affect the puff loss emissions in Japan. Figure 6c indicates that the puff loss emissions were affected by gasoline type (or gasoline RVP) and season (or ambient temperature). In general, WGG is manufactured from October to April while SGG is manufactured in the remaining seasons in Japan [28]. Notably, a mixture of WGG and SGG

is sold during the transition periods (October and May). The estimated puff loss emissions were similar in July 2019 (SGG), December 2019 (WGG), and April 2020 (WGG). Thus, the seasonal effects on puff loss emissions are expected to be stable under constant ambient pressure conditions. Finally, according to Figure 6c, the puff loss emissions per refueling event from the tested vehicle in this study approximately ranged from 2 to 5 g. The annual average driving distance of passenger vehicles in Japan is approximately 10,000 km, based on a statistical study [29], and the fuel consumption of the tested vehicle is 11.2 L/100 km based on the worldwide harmonized light duty driving test cycle (WLTC), measured via the chassis dynamometer test (details on this test are described in one of our previous studies [25]). The size of the fuel tank is 70 L, and we assumed that refueling is done when the amount of fuel in the tank drops to 20 L. The number of gasoline vehicles in Japan is approximately 60 million [30]. From this information, assuming that all gasoline vehicle properties are the same as those of the tested vehicle in this study, the annual total puff loss emissions in Japan are roughly calculated to range from 2700 to 6700 t/y. The total amount of VOC emissions from stationary sources in Japan, 2019 was 640,000 t/y [31], so that the puff loss emissions account for 0.42% to 1.05% of the VOC emissions from stationary sources. The Japanese government has conducted a rational VOC reduction management since 2004, the so called "best mix policy", and the amount of VOC emissions from stationary sources has been gradually decreased after its commencement. The importance of managing puff loss emissions as a VOC emission source is important to further implement a VOC reduction policy in the near future.

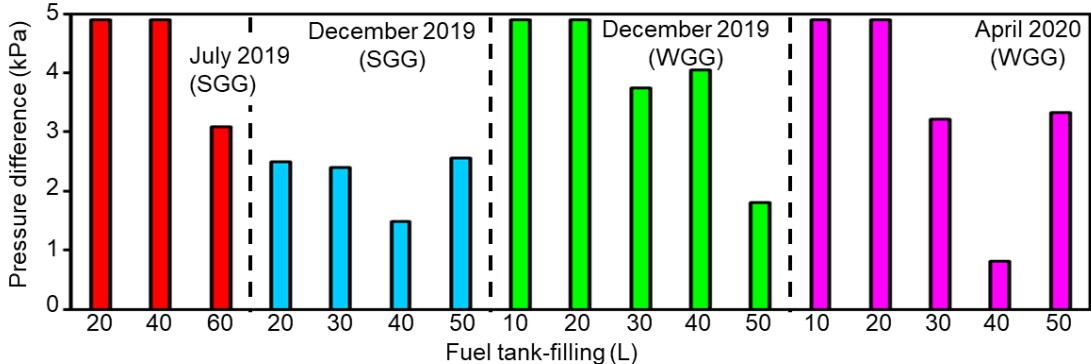

**Figure 5.** Calculated pressure difference between the interior and exterior of the fuel tank under different experimental conditions.

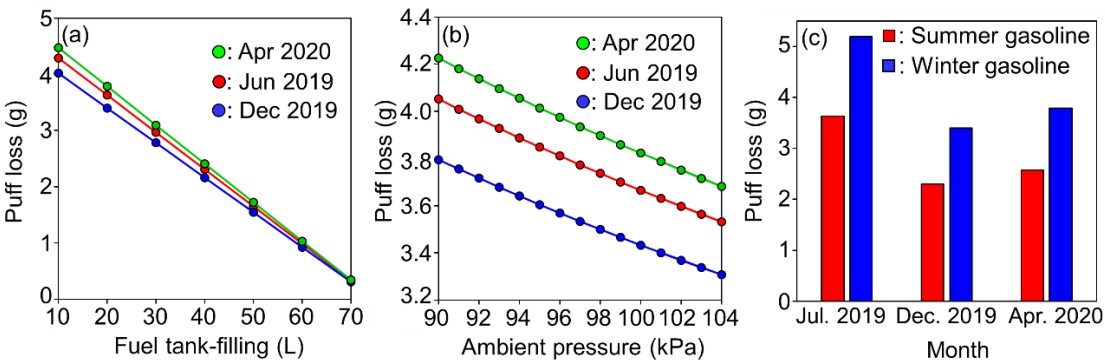

**Figure 6.** The sensitivity of puff loss emissions to (**a**) fuel tank-filling, (**b**) ambient pressure, and (**c**) fuel type (Reid vapor pressure (RVP)$_{SGG}$: 59 kPa; RVP$_{WGG}$: 88 kPa).

## 5. Conclusions

In this study, the behavior of puff loss emissions was investigated in response to different seasons and gasoline types, i.e., July 2019 (SGG), December 2019 (SGG), December 2019 (WGG), and April 2020

(WGG), in one gasoline vehicle. The puff loss emissions were trapped by an activated carbon canister. Subsequently, their quantity was determined as the weight difference of the activated carbon canister before and after VOC adsorption. A detailed composition analysis of the puff loss emissions was conducted using GC-MS/FID. The experimental results indicated the seasonal dependency of the quantity of puff loss emissions. Specifically, the emissions were the highest in July 2019, owing to the high summer temperatures. Overall, the emissions indicated complex relationships with physical parameters such as ambient temperature, ambient pressure, gasoline RVP, and fuel tank-filling. Subsequently, we constructed a theoretical model to investigate these complex trends. In this regard, the theory of thermodynamics was utilized to determine the quantity of puff loss emissions under arbitrary environmental conditions. The results indicated good agreement between the measured and calculated puff loss emissions. Finally, a sensitivity analysis was conducted for the puff loss estimation model. The results of the sensitivity analysis indicated that fuel tank-filling was the most important factor affecting the quantity of puff loss emissions. However, this is the first study in which the measurement and quantification of puff loss emissions indicated a constant VOC emission owing to fuel-cap removal regardless of the season; the experimental values were obtained for a single gasoline vehicle. The Japanese government has been implementing a policy to reduce VOC emissions from the refueling process based on Stage II technology, but this method does not include puff loss emissions, causing an underestimation of the total VOC emissions. Therefore, it is important to validate the proposed theoretical model for different gasoline vehicles. Thus, we plan to conduct puff loss experiments on several other vehicles in the near future. The validated theoretical model is likely to aid the evaluation of VOC emission inventories. Subsequently, these results can be incorporated into chemical transport models to determine the atmospheric concentrations of tropospheric ozone and $PM_{2.5}$.

**Supplementary Materials:** The following is available online at http://www.mdpi.com/2073-4433/11/10/1110/s1, Table S1: Parameters of the thermodynamic model, Table S2: Measured values of puff loss emissions, Table S3: Measured values of ambient temperatures, Table S4: Measured values of fuel temperatures, Table S5: Measured values of fuel-tank's empty space temperatures.

**Author Contributions:** K.T. contributed to the research design. H.H., S.-y.T., H.Y., and K.T. planned the experiments. H.H., S.-y.T., and K.T. conducted the experiments. H.H., S.-y.T., and G.N. proposed the theoretical model. S.-y.T. and G.N. created the calculation program of the theoretical model. H.H., S.-y.T., G.N., H.Y., and K.T. wrote the paper. All authors have read and agreed to the published version of the manuscript.

**Funding:** A part of this study was funded by the Ministry of the Environment Government of Japan. The APC was funded by TMRIEP.

**Acknowledgments:** We are grateful to Koichi Yanai and Kiyoshi Yanami from the Tokyo Metropolitan Research Institute for Environmental Protection (TMRIEP) and Yasuyuki Shirai from the Tokyo Denki University for their assistance in the experimental setup. We are also grateful to Megumi Okada from TMRIEP for assisting with the GC-MS/FID-based VOC analysis.

**Conflicts of Interest:** The authors declare no conflict of interest.

## Appendix A

**Table A1.** Specifications of the tested vehicle.

| Manufacturer | Toyota |
| --- | --- |
| Vehicle type | Cargo |
| Engine | Regular gasoline port injection with in-line four-cylinder |
| Vehicle weight (kg) | 1760 |
| Displacement (L) | 1.998 |
| Fuel tank volume (L) | 70 |

**Table A2.** The composition of summer grade gasoline used in July 2019 (vol%).

| Carbon Number | *n*-Alkane | *iso*-Alkane | Alkenes | Naphthene | Aromatics | Total |
|---|---|---|---|---|---|---|
| C3 | 0.02 | - | 0.00 | - | - | 0.02 |
| C4 | 1.87 | 0.81 | 0.83 | - | - | 3.51 |
| C5 | 5.82 | 9.80 | 4.73 | 0.37 | - | 20.72 |
| C6 | 4.56 | 9.73 | 3.45 | 2.32 | 0.50 | 20.56 |
| C7 | 1.61 | 6.96 | 2.67 | 2.23 | 6.53 | 20.00 |
| C8 | 0.47 | 3.15 | 1.63 | 1.42 | 5.19 | 11.86 |
| C9 | 0.19 | 1.88 | 0.49 | 0.77 | 5.93 | 9.26 |
| C10 | 0.13 | 1.24 | 0.38 | 0.17 | 2.92 | 4.84 |
| C11 | 0.10 | 0.62 | 0.14 | 0.07 | 1.03 | 1.96 |
| C12 | 0.04 | 0.31 | 0.05 | 0.00 | 0.16 | 0.56 |
| C13 | 0.02 | 0.02 | 0.00 | 0.00 | 0.00 | 0.04 |
| Total | 14.83 | 34.52 | 14.37 | 7.35 | 22.26 | 93.33 |

**Table A3.** The composition of summer grade gasoline used in December 2019 (vol%).

| Carbon Number | *n*-Alkane | *iso*-Alkane | Alkenes | Naphthene | Aromatics | Total |
|---|---|---|---|---|---|---|
| C3 | 0.02 | - | 0.00 | - | - | 0.02 |
| C4 | 2.05 | 0.92 | 0.99 | - | - | 3.96 |
| C5 | 4.50 | 8.55 | 4.38 | 0.28 | - | 17.71 |
| C6 | 4.67 | 11.97 | 3.39 | 2.21 | 0.45 | 22.69 |
| C7 | 2.06 | 8.98 | 2.26 | 2.04 | 7.87 | 23.21 |
| C8 | 0.46 | 2.74 | 2.77 | 1.33 | 5.48 | 12.78 |
| C9 | 0.16 | 1.55 | 0.59 | 0.73 | 5.59 | 8.62 |
| C10 | 0.10 | 1.05 | 0.33 | 0.13 | 2.02 | 3.63 |
| C11 | 0.06 | 0.53 | 0.11 | 0.05 | 0.54 | 1.29 |
| C12 | 0.02 | 0.12 | 0.05 | 0.00 | 0.05 | 0.24 |
| C13 | 0.01 | 0.00 | 0.00 | 0.00 | 0.00 | 0.01 |
| Total | 14.11 | 36.41 | 14.87 | 6.77 | 22.00 | 94.16 |

**Table A4.** The composition of winter grade gasoline used in December 2019 (vol%).

| Carbon Number | *n*-Alkane | *iso*-Alkane | Alkenes | Naphthene | Aromatics | Total |
|---|---|---|---|---|---|---|
| C3 | 0.07 | - | 0.01 | - | - | 0.08 |
| C4 | 4.05 | 3.33 | 2.62 | - | - | 10.00 |
| C5 | 4.97 | 9.17 | 4.71 | 0.66 | - | 19.51 |
| C6 | 4.16 | 10.84 | 3.24 | 2.35 | 0.50 | 21.09 |
| C7 | 1.55 | 7.16 | 2.13 | 1.99 | 6.45 | 19.28 |
| C8 | 0.42 | 2.66 | 2.32 | 1.27 | 3.84 | 10.51 |
| C9 | 0.16 | 1.66 | 0.50 | 0.64 | 5.04 | 8.00 |
| C10 | 0.10 | 1.05 | 0.28 | 0.12 | 2.00 | 3.55 |
| C11 | 0.06 | 0.55 | 0.10 | 0.05 | 0.48 | 1.24 |
| C12 | 0.02 | 0.12 | 0.12 | 0.00 | 0.05 | 0.31 |
| C13 | 0.00 | 0.00 | 0.00 | 0.00 | 0.00 | 0.00 |
| Total | 15.56 | 36.54 | 16.03 | 7.08 | 18.36 | 93.57 |

**Table A5.** Composition analysis of puff loss emissions by the GC-MS/FID (vol%).

| Organic Compound | Experimental Month | | | |
|---|---|---|---|---|
| | July 2019 (SGG) [1] | December 2019 (SGG) [1] | December 2019 (WGG) [2] | April 2020 (WGG) [2] |
| ethane | 0.40 | 0.13 | 0.09 | 0.26 |
| ethylene | 0.64 | 0.00 | 0.00 | 0.15 |
| propane | 0.23 | 1.22 | 1.50 | 1.16 |
| propylene | 0.00 | 0.08 | 0.22 | 0.24 |
| acetylene | 0.00 | 0.00 | 0.00 | 0.12 |
| isobutane | 4.80 | 13.03 | 26.18 | 22.07 |
| n-butane | 8.15 | 17.04 | 19.65 | 16.63 |
| trans-2-butene | 2.13 | 5.66 | 11.52 | 7.97 |

**Table A5.** *Cont.*

| Organic Compound | Experimental Month | | | |
|---|---|---|---|---|
| | July 2019 (SGG) [1] | December 2019 (SGG) [1] | December 2019 (WGG) [2] | April 2020 (WGG) [2] |
| 1-butene | 0.46 | 1.32 | 1.24 | 1.50 |
| isobutene | 0.58 | 1.42 | 0.63 | 1.51 |
| cis-2-butene | 1.46 | 3.79 | 5.75 | 4.09 |
| isopentane | 24.16 | 26.07 | 18.52 | 15.42 |
| n-pentane | 10.76 | 7.25 | 4.25 | 6.91 |
| trans-2-pentene | 1.21 | 0.80 | 0.46 | 0.72 |
| 1,3-butadiene | 0.68 | 0.00 | 0.00 | 0.00 |
| 3-methyl-1-butene | 0.54 | 0.57 | 0.37 | 0.27 |
| 1-pentene | 1.66 | 0.75 | 0.42 | 0.56 |
| 2-methyl-1-butene | 0.72 | 1.56 | 0.89 | 1.17 |
| cis-2-pentene? | 1.82 | 1.39 | 0.79 | 1.37 |
| 2-methyl-2-butene | 2.56 | 1.93 | 1.22 | 2.09 |
| isoprene | 0.00 | 0.00 | 0.00 | 0.00 |
| cis-1,3-pentadiene | 0.00 | 0.00 | 0.00 | 0.05 |
| t-1,3-pentadiene | 0.00 | 0.00 | 0.00 | 0.01 |
| 2,2-dimethylbutane | 0.79 | 0.64 | 0.31 | 0.33 |
| cyclopentane | 0.84 | 0.43 | 0.42 | 0.93 |
| 2,3-dimethylbutane | 0.80 | 0.93 | 0.48 | 0.82 |
| 2-methylpentane | 4.76 | 3.09 | 1.38 | 2.60 |
| 3-methylpentane | 3.56 | 2.19 | 0.89 | 1.82 |
| 2-methyl-1-pentene | 0.10 | 0.11 | 0.05 | 0.11 |
| 1-hexene | 0.00 | 0.12 | 0.05 | 0.10 |
| hexane | 3.85 | 1.69 | 0.67 | 1.63 |
| cis-3-hexene | 0.07 | 0.10 | 0.04 | 0.04 |
| cis-2-hexene | 0.73 | 0.25 | 0.11 | 0.15 |
| cis-3-methyl-2-pentene | 0.26 | 0.14 | 0.07 | 0.10 |
| t-2-hexene | 0.40 | 0.17 | 0.07 | 0.21 |
| ethyl-tert-butylether | 6.22 | 1.74 | 0.34 | 2.13 |
| t-3-methyl-2-pentene | 0.29 | 0.18 | 0.08 | 0.16 |
| methylcyclopentane | 1.83 | 0.82 | 0.38 | 1.08 |
| 2,4-dimethylpentane | 0.56 | 0.14 | 0.04 | 0.16 |
| benzene | 0.69 | 0.18 | 0.09 | 0.30 |
| cyclohexane | 0.38 | 0.13 | 0.07 | 0.19 |
| 2-methylhexane | 1.73 | 0.63 | 0.17 | 0.68 |
| 2,3-dimethylpentane | 0.55 | 0.18 | 0.05 | 0.16 |
| 3-methylhexane | 1.23 | 0.57 | 0.16 | 0.54 |
| 1-heptene | 0.34 | 0.11 | 0.04 | 0.12 |
| 2,2,4-trimethylpentane | 0.01 | 0.00 | 0.00 | 0.02 |
| heptane | 0.82 | 0.25 | 0.05 | 0.20 |
| methylcyclohexane | 0.40 | 0.07 | 0.02 | 0.08 |
| 2,3,4-trimethylpentane | 0.15 | 0.00 | 0.00 | 0.00 |
| toluene | 3.48 | 0.84 | 0.23 | 0.81 |
| 2-methylheptane | 0.11 | 0.04 | 0.01 | 0.03 |
| 3-methylheptane | 0.19 | 0.04 | 0.01 | 0.03 |
| octane | 0.03 | 0.01 | 0.00 | 0.01 |
| ethylbenzene | 0.34 | 0.04 | 0.01 | 0.03 |
| m-xylene | 0.60 | 0.07 | 0.01 | 0.05 |
| p-xylene | 0.16 | 0.02 | 0.00 | 0.02 |
| styrene | 0.00 | 0.00 | 0.00 | 0.00 |
| o-xylene | 0.22 | 0.03 | 0.00 | 0.02 |
| nonane | 0.03 | 0.00 | 0.00 | 0.00 |
| isopropylbenzene | 0.02 | 0.00 | 0.00 | 0.00 |
| alpha-pinene | 0.00 | 0.00 | 0.00 | 0.00 |
| propylbenzene | 0.02 | 0.01 | 0.00 | 0.00 |
| m-ethyltoluene | 0.13 | 0.01 | 0.00 | 0.01 |
| p-ethyltoluene | 0.00 | 0.01 | 0.00 | 0.00 |
| 1,3,5-trimethylbenzene | 0.02 | 0.01 | 0.00 | 0.00 |
| o-ethyltoluene | 0.03 | 0.00 | 0.00 | 0.00 |
| beta-pinene | 0.00 | 0.00 | 0.00 | 0.00 |
| 1,2,4-trimethylbenzene | 0.11 | 0.02 | 0.00 | 0.01 |
| decane | 0.03 | 0.00 | 0.00 | 0.00 |
| 1,2,3-trimethylbenzene | 0.00 | 0.00 | 0.00 | 0.00 |
| m-diethylbenzene | 0.00 | 0.00 | 0.00 | 0.00 |
| p-diethylbenzene | 0.00 | 0.00 | 0.00 | 0.00 |
| 2-ethyl-p-xylene | 0.05 | 0.00 | 0.00 | 0.00 |
| 4-ethyl-m-xylene | 0.04 | 0.00 | 0.00 | 0.00 |
| undecane | 0.07 | 0.01 | 0.00 | 0.01 |
| 1,2,3,5-tetramethylbenzene | 0.02 | 0.00 | 0.00 | 0.00 |

[1] SGG: Summer grade gasoline. [2] WGG: Winter grade gasoline.

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
