# Peer review of "Evaluation of Gasoline Evaporative Emissions from Fuel-Cap Removal after a Real-World Driving Event"

_atmosphere, doi:10.3390/atmos11101110_

Round 1

Reviewer 1 Report

Dear Editor, 

Thank you for sending me the revised manuscript

After revision, this work had been significantly improved. It is meaningful to evaluate the gasoline evaporative emissions from fuel-cap removal for a typical gasoline vehicle. This work will also aid the evaluation of future VOC emissions inventories.

Thus, may I suggest 'accept' by the journal.

Author Response

The authors feel gratitude to spend a time as a reviewer for our manuscript. Thank you for the decision for acceptance.

Reviewer 2 Report

Dear Authors

The paper studies the "puff loss" issue in gasoline refilling with experimental and estimation methods. The idea is mindblowing and undoubtedly important. The methodological execution of the analyses are fine, albeit the results are not very surprising. I have the following comments on the paper:

  • Introduction: You might discuss more deeply about the malignancy of the issue. If it is easily done, you might give some rough estimates about daily puff losses in Tokyo (daily tank openings times the average puff loss), about the contents of the noxious vapor inhaled by the refuelers themselves, and about the gains in energy efficiency and environmental effects in the alternative (catalysis assisted) tailpipe mode. That is, you could motivate the paper as an urge to cut puff losses in order to protect the drivers and meet the environmental standards. You might even reconsider your research question: instead of highlighting the atmospheric problem, you might focus on the micro-atmosphere in the gas stations.  
  • Methodology: Fine. The tank ventilation technique (rows 191-4) might be worth of more discussion because it's a twin-edged sword - it exhalates emissions but saves the refueler from inhaling that part of them. Thus, a low max. pressure is bad for atmosphere in general, but good for the driver's micro-atmosphere.   
  • Results and discussion: Fine, but one might expect closer scrutiny on the findings, like concerning the low values in Fig 1, Dec SGG. Low volatility fuel in the wintertime sounds good for both micro- and macro-atmosphere, but the starting problems etc. may lower energy efficiency, or what? One may also wonder why Table S2 is partly in Finnish. 
  • Puff loss estimation model: Fine, but some further discussion would be illuminative.
  • Conclusions: Particularly if you decide to reset your research question, your results concerning the essence of temperature and tank filling should be apt for policy implications. These could concern regulations for car manufacturers (tank material and cooling, inner rubber bags to prevent vaporization etc.), instructions for gas stations (warning signs, ventilation of the gas pump field etc.), and ways of monitoring (for example on-site measurement of emissions etc.).        

Author Response

Reviewer 2:

Comments and Suggestions for Authors:

Dear Authors

The paper studies the "puff loss" issue in gasoline refilling with experimental and estimation methods. The idea is mindblowing and undoubtedly important. The methodological execution of the analyses are fine, albeit the results are not very surprising. I have the following comments on the paper:

The authors are grateful for the positive opinion and important suggestions for out manuscript. The manuscript was modified and improved based on the suggestion.

Introduction: You might discuss more deeply about the malignancy of the issue. If it is easily done, you might give some rough estimates about daily puff losses in Tokyo (daily tank openings times the average puff loss), about the contents of the noxious vapor inhaled by the refuelers themselves, and about the gains in energy efficiency and environmental effects in the alternative (catalysis assisted) tailpipe mode. That is, you could motivate the paper as an urge to cut puff losses in order to protect the drivers and meet the environmental standards. You might even reconsider your research question: instead of highlighting the atmospheric problem, you might focus on the micro-atmosphere in the gas stations.

Thank you for the meaningful suggestion. To our best knowledge, the amount of puff loss emission has never been evaluated in worldwide, and thus, it is difficult to add the related reference in Introduction section. Despite this, we found it is necessary about the importance of including roughly total amount of puff loss emission to convince the policy makers or researchers to conduct further study of puff loss emissions. Therefore, the roughly total amount of puff loss emission in Japan was added in the estimation model section as following.

Lines 317-333: “Finally, according to Figure 6(c), the puff loss emissions per refueling event from the tested vehicle in this study approximately ranged from 2 to 5 g. The annual average driving distance of passenger vehicles in Japan is approximately 10,000 km, based on a statistical study [29], and the fuel consumption of the tested vehicle is 11.2 L/100 km based on the worldwide harmonized light duty driving test cycle (WLTC), measured via the chassis dynamometer test (details on this test are described in one of our previous studies [25]). The size of the fuel tank is 70 L, and we assumed that refueling is done when the amount of fuel in the tank drops to 20 L. The number of gasoline vehicles in Japan is approximately 60 million [30]. From this information, assuming that all gasoline vehicle properties are the same as those of the tested vehicle in this study, the annual total puff loss emissions in Japan are roughly calculated to range from 2,700 t/y to 6,700 t/y. The total amount of VOC emissions from stationary sources in Japan, 2019 was 640,000 t/y [31], so that the puff loss emissions account for 0.42 % to 1.05 % of the VOC emissions from stationary sources. The Japanese government has conducted a rational VOC reduction management since 2004, so called “best mix policy”, and the amount of VOC emissions from stationary sources has been gradually decreased after its commencement. The importance of managing puff loss emissions as a VOC emission source is important to further implement the VOC reduction policy in the near future.”

Methodology: Fine. The tank ventilation technique (rows 191-4) might be worth of more discussion because it's a twin-edged sword - it exhalates emissions but saves the refueler from inhaling that part of them. Thus, a low max. pressure is bad for atmosphere in general, but good for the driver's micro-atmosphere.

Thank you for the suggestion about maximum tank inner pressure. According to the advice, following sentence was added in the manuscript.

Lines 194-196: “, and there is a recent trend of an increasing maximum pressure in fuel tanks to prevent VOC emissions to the air and inhaled by humans in the refueling process.”

Results and discussion: Fine, but one might expect closer scrutiny on the findings, like concerning the low values in Fig 1, Dec SGG. Low volatility fuel in the wintertime sounds good for both micro- and macro-atmosphere, but the starting problems etc. may lower energy efficiency, or what? One may also wonder why Table S2 is partly in Finnish.

According to the meaningful suggestion, the following sentence was added in the revised manuscript.

Lines 205-207: “Despite this fact, using low volatility fuel at ambient temperatures would lead to a low combustion efficiency of the engine system and, therefore, the combination of low volatility fuel with low ambient temperatures is not recommended.”

The notation of the date was changed from Finnish to American.

Puff loss estimation model: Fine, but some further discussion would be illuminative.

According to the earlier suggestion of the Introduction, total amount of puff loss emissions in whole year of Japan, 2015 was roughly estimated and added in this section.

Conclusions: Particularly if you decide to reset your research question, your results concerning the essence of temperature and tank filling should be apt for policy implications. These could concern regulations for car manufacturers (tank material and cooling, inner rubber bags to prevent vaporization etc.), instructions for gas stations (warning signs, ventilation of the gas pump field etc.), and ways of monitoring (for example on-site measurement of emissions etc.).

Thank you for the suggestions regarding our conclusions. Since the focus of this study is atmospheric pollution and the evaluation of emission inventory from puff loss, we decided not including the policy implications related to vehicle manufacturer issue. Nevertheless, we were convinced about the reviewer’s opinion, and the following sentence was added which is related to the policy implication of Stage 2, the policy of refueling emission.

This manuscript is a resubmission of an earlier submission. The following is a list of the peer review reports and author responses from that submission.

Round 1

Reviewer 1 Report

Thank you for sending me the attached manuscript for review.

This study evaluated gasoline evaporative emissions from fuel-cap removal during the 14 refueling process (or “puff loss”) for one gasoline vehicle in the Japanese market. The authors developed an estimation model to determine puff loss emissions under arbitrary environmental conditions.  

It is meaningful to discuss the measurements and removal of VOCs and PM2.5 emissions of different vehicles from real-world driving. This work may contribute to the VOCs emission factors and inventory from vehicles. The manuscript is well-written and structured. The language if fine.

My detailed comments are listed below:

  1. In the abstract, it is suggested to add some results obtained from this work.
  2. In the introduction section, the authors only focus on the literature in Japan. Aerosol emissions and their control technologies from developing countries also from Asia, for example, China, is suggested to be mentioned and discussed.

https://www.sciencedirect.com/science/article/abs/pii/S1352231017308701

https://www.sciencedirect.com/science/article/abs/pii/S0959652620332339

3. Section 4.1, line 247-261, the model results are suggested to be further introduced and discussed, for example, what is the error of the regression model? For example, SEM? And why there are some significant outliers in figure 4?

4. Did the authors compare their results with previous research? It is suggested that the authors compare their results with literature.

5. In the conclusion section, the authors are suggested to be extended with some discussions on what are the further implications of this work.

Author Response

Thank you for sending me the attached manuscript for review.

This study evaluated gasoline evaporative emissions from fuel-cap removal during the 14 refueling process (or “puff loss”) for one gasoline vehicle in the Japanese market. The authors developed an estimation model to determine puff loss emissions under arbitrary environmental conditions.

It is meaningful to discuss the measurements and removal of VOCs and PM2.5 emissions of different vehicles from real-world driving. This work may contribute to the VOCs emission factors and inventory from vehicles. The manuscript is well-written and structured. The language if fine.

The authors feel gratitude for the positive response and valuable suggestions to our study. The manuscript was modified in accordance with the reviewer’s suggestions. The language was modified by the professional proofreader.

My detailed comments are listed below:

In the abstract, it is suggested to add some results obtained from this work.

Thank you for the suggestion. Following sentence was added in the abstract to clarify the concrete amount of puff loss emission from the tested vehicle.

Line 19: “This higher puff loss emissions accounted maximally for more than 4 g of the emissions from the tested vehicle.”

In the introduction section, the authors only focus on the literature in Japan. Aerosol emissions and their control technologies from developing countries also from Asia, for example, China, is suggested to be mentioned and discussed.

https://www.sciencedirect.com/science/article/abs/pii/S1352231017308701

https://www.sciencedirect.com/science/article/abs/pii/S0959652620332339

According to the suggestion, the other countries situation including U.S., E.U., and China were added as following with the suggested references.

Lines 45-50: “While developed countries such as the United States and European Union countries have air quality levels similar to that in Japan [8,9], middle- and low-income countries including China are still suffering from extreme air pollution and carbon emission, and managing air quality based on the careful planning is still challenging [10,11]. Therefore, for both developed and developing countries, it is necessary to develop mitigation strategies for air pollutants, including ozone precursors.”

  1. Section 4.1, line 247-261, the model results are suggested to be further introduced and discussed, for example, what is the error of the regression model? For example, SEM? And why there are some significant outliers in figure 4?

Thank you for pointing out the lack of the information about the model error. The error was added by the form of route mean square error (RMSE) in Figure 4. The error of the plot in ~2 g experimental result was added by the following sentence.

Lines 258-263: “Since we did not record the exact duration of the experiment and could not determine the exact ambient pressure for each condition, the ambient pressures at 12 p.m. listed in Table 1 were applied in the calculations. The duration of each test was 6 h, from 9:00 a.m. to 3:00 p.m., and according to data provided by the Japan Meteorological Agency, there were no drastic changes in ambient pressure [24]. Therefore, the use of ambient pressure at 12 p.m. in the calculations did not lead to considerable errors.”

  1. Did the authors compare their results with previous research? It is suggested that the authors compare their results with literature.

To the best of our knowledge, this is the first study for the experiment and modeling of the evaporation from puff loss, and therefore, the comparison of the results to the previous study was not conducted. This is mentioned in the lines 63-64 as the introduction.

  1. In the conclusion section, the authors are suggested to be extended with some discussions on what are the further implications of this work.

According to the suggestion, the possible indication of this study was added as the following sentence.

Lines 315-317: “However, this is the first study in which the measurement and quantification of puff loss emissions indicated a constant VOC emission owing to fuel cap removal regardless of the season;”

Reviewer 2 Report

The authors reported the results of a study on puff loss emissions from gasoline vehicles. The puff loss emissions from gasoline vehicles are important. This is the first reported study on this topic. The manuscript has a merit. However, the paper needs improvements on data presentations and more data collections.

On Page 5, the authors reported measured results by a figure (Figure 1). Since this is the first study on this topic, it is better to report raw data (testing dates, temperatures (ambient and tank), pressures, and measured weight increasing for each test, etc.). Those data will help other scientists understand the results and explanations better.

It is not clear whether the author did any repeating tests for each or some testing conditions. In order to understand trends, especially when the authors did not get consistent results, the test method precisions are important to know.

The explanations of the result of tests on SGG December 2019 is not convinced as well as that for the data of 10 L WGG April 2020. Raw data and test precision data will help readers to make their own judgements.    

Author Response

The authors reported the results of a study on puff loss emissions from gasoline vehicles. The puff loss emissions from gasoline vehicles are important. This is the first reported study on this topic. The manuscript has a merit. However, the paper needs improvements on data presentations and more data collections.

The authors feel gratitude for the positive opinion and valuable suggestions for our manuscript. The manuscript was carefully modified based on the suggestions from the reviewers. Raw data of the experiments are supplied as the supplementary material.

On Page 5, the authors reported measured results by a figure (Figure 1). Since this is the first study on this topic, it is better to report raw data (testing dates, temperatures (ambient and tank), pressures, and measured weight increasing for each test, etc.). Those data will help other scientists understand the results and explanations better.

Thank you for the suggestion. The raw data from the experiments were added as the supplementary material.

It is not clear whether the author did any repeating tests for each or some testing conditions. In order to understand trends, especially when the authors did not get consistent results, the test method precisions are important to know.

We did not conduct the repeating tests in this study because the ambient conditions (e.g. temperature, ambient pressure, weather) and traffic condition were different day by day and it is difficult to gain the same experimental condition from the real-world experiment. Despite this fact, the proposed model to evaluate the amount of puff loss emissions well fit to the experimental results even some errors were observed, and the possible reasons of the error was described in the manuscript. As described in the main article, we conducted the experiments only for one vehicle, so further study should be done in the near future of which the authors have already planned to the next observations. The purpose of this study is to provide and prompt the motivation for the research of puff loss emissions for the researchers (or policy makers) worldwide.

The explanations of the result of tests on SGG December 2019 is not convinced as well as that for the data of 10 L WGG April 2020. Raw data and test precision data will help readers to make their own judgements.

The behavior of the evaporation from gasoline vehicles is strongly affected by RVP and fuel temperature (or the difference of the temperatures before and after the vehicle running), which is indicated by our previous study for parking evaporation (Hata et al. Sci. Total Environ. 2018: https://doi.org/10.1016/j.scitotenv.2017.10.030). The tests for SGG on December 2019 were conducted with low RVP fuel in low temperature condition, resulted in the low evaporation. On the other hand, the tests for WGG on April 2020 were in the high RVP fuel in relatively high temperature condition, getting the high amount of evaporation. Meanwhile, the raw data were added accounting for the suggestion to be checked by the future readers.
